# AUE-Net: Automated Generation of Ultrasound Elastography Using Generative Adversarial Network

**DOI:** 10.3390/diagnostics12020253

**Published:** 2022-01-20

**Authors:** Qingjie Zhang, Junjuan Zhao, Xiangmeng Long, Quanyong Luo, Ren Wang, Xuehai Ding, Chentian Shen

**Affiliations:** 1School of Computer Engineering and Science, Shanghai University, Shanghai 200444, China; ivanzhang@shu.edu.cn (Q.Z.); junjuanzhao@shu.edu.cn (J.Z.); matsuko@shu.edu.cn (X.L.); 2Department of Nuclear Medicine, Shanghai Jiao Tong University Affiliated Sixth People’s Hospital, Shanghai 200233, China; lqyn@sh163.net; 3Department of Ultrasound Medicine, Shanghai Jiao Tong University Affiliated Sixth People’s Hospital, Shanghai 200233, China; ryanwang@126.com

**Keywords:** ultrasound elastography, neural networks, generative adversarial networks, attention module, color loss

## Abstract

Problem: Ultrasonography is recommended as the first choice for evaluation of thyroid nodules, however, conventional ultrasound features may not be able to adequately predict malignancy. Ultrasound elastography, adjunct to conventional B-mode ultrasound, can effectively improve the diagnostic accuracy of thyroid nodules. However, this technology requires professional elastography equipment and experienced physicians. Aim: in the field of computational medicine, Generative Adversarial Networks (GANs) were proven to be a powerful tool for generating high-quality images. This work therefore utilizes GANs to generate ultrasound elastography images. Methods: this paper proposes a new automated generation method of ultrasound elastography (AUE-net) to generate elastography images from conventional ultrasound images. The AUE-net was based on the U-Net architecture and optimized by attention modules and feature residual blocks, which could improve the adaptability of feature extraction for nodules of different sizes. The additional color loss function was used to balance color distribution. In this network, we first attempted to extract the tissue features of the ultrasound image in the latent space, then converted the attributes by modeling the strain, and finally reconstructed them into the corresponding elastography image. Results: a total of 726 thyroid ultrasound elastography images with corresponding conventional images from 397 patients were obtained between 2019 and 2021 as the dataset (646 in training set and 80 in testing set). The mean rating accuracy of the AUE-net generated elastography images by ultrasound specialists was 84.38%. Compared with that of the existing models in the visual aspect, the presented model generated relatively higher quality elastography images. Conclusion: the AUE-net generated ultrasound elastography images showed natural appearance and retained tissue information. Accordingly, it seems that B-mode ultrasound harbors information that can link to tissue elasticity. This study may pave the way to generate ultrasound elastography images readily without the need for professional equipment.

## 1. Introduction

The superficial location of the thyroid gland makes high-resolution ultrasound the imaging modality of choice for the evaluation of diffuse disease and nodules. Patients may be referred for ultrasonography because of palpable abnormalities. However, detection of nonpalpable nodules can be as high as 70% and varies by imaging modality [1]. In general, the prevalence of thyroid nodules is high and they may be neoplastic. Most of nodules are benign proliferative structural changes and benign follicular adenomas, with only a small percentage of malignancy [2].

Currently, further confirmation of malignancy for thyroid nodulemainly relies on Fine Needle Aspiration Cytology (FNAC). But given the high prevalence of thyroid nodules and the fact that most nodules are benign, too many unnecessary FNAC operations were possibly conducted, which results in medical resources wasting. Studies [3,4,5,6] showed that tissue elasticity is positively correlated with the likelihood of malignancy. This stimulated considerable progress in the development of a new technique, elastography, in the last decades. Strain elastography (SE), the first introduced ultrasound elastography, is a promising new tool to effectively improve diagnostic accuracy and reduce medical costs. In the last decades, SE combined with conventional ultrasound and FNAC became an important strategy for differentiating benign and malignant thyroid nodules [7].

SE is the most widely used commercial method for direct stiffness assessment during thyroid examinations, but it requires an add-on module to be combined with a conventional ultrasound transducer [8]. In addition, elastography also relies on physicians’ experience. Different specialists’ skillsets may cause variable tissue deformation from compression. It is difficult to standardize the pressure and strain variations. Nonuniform compression produces intra-observer and interobserver variability [4]. Extensive training and experience are crucial for obtaining reliable and reproducible elastography images and scores. For elastography scoring, the main criterion is the stiffness ratio between the hard area (e.g., blue area) and the nodule area. The second scoring element is the color distribution. These two features correspond to the spatial location relationship and the RGB three-channel relationship in computer image processing. This provides the prerequisite for using computer imaging algorithms to generate elastography images.

In the last decade of computing science, deep learning was gaining monumental traction, with neural networks leading the way. Numerous research works made important contributions in fields, such as the classification of lung and breast diseases [9,10] and the identification and segmentation of gynecological abnormality [11,12]. Since Goodfellow and others proposed generative adversarial networks (GANs) in 2014 [13], GANs were shown a promising future in building image synthesis and data generation techniques. There is an increasing amount of researches on GANs, with each iteration making further progress [14,15]. Due to the excellent performance of GANs on natural image synthesis, GANs were introduced to perform various medical imaging tasks including solving problems associated with data imbalance of virtual STIR images [16], reducing metal artifacts and the radiation dose during digital tomosynthesis [17], as well as helping to process medical image [18,19]. Although neural networks and GANs were used to extract strain images from radio frequency data [20,21], and generate shear wave elastography images [22], we attempted to directly map conventional ultrasound images towards the corresponding strain elastography (SE) images. To the best of our knowledge, this is the first work to apply the nonphysical method to generate strain elastography images from conventional ultrasound images.

The main contributions of this paper are as follows: (1) We introduced GAN networks to thyroid strain elastography image generation. (2) We proposed a whole framework for elastography image generation from raw ultrasound data. We used a preprocessing method to get image pairs from raw data and a conditional GAN-based automated generating network (AUE-net) to generate elastography images. (3) This method can generate elastography images corresponding to thyroid ultrasound images only through nonphysical computer algorithms. In this way, the need for professional equipment and manual intervention will be reduced.

The structure of this paper is organized as following. Section 1 introduces relevant research background and our study. In Section 2, we first introduce the base generative adversarial networks. Second, the datasets, network architecture, experiment details, and evaluation metrics are presented. Section 3 presents the results of the USE-GAN method in fields of visual comparisons and quantitative metrics. Lastly, Section 4 details the conclusion.

## 2. Materials and Methods

### 2.1. Genaraitve Adversarial Networks and Pix2pix

Since Goodfellow and others proposed GAN in 2014 [13], many studies explored very important paths using GANs in a variety of medical application scenarios. Bermudez and other researchers [18] trained a GAN to synthesize new T1-weighted brain MRIs with high quality compared to that of real images. In the paper by Gomi and others [17], the authors showed that GAN-based networks could achieve promising results in LDCT denoising. GANs were also used to generate more training data. Shin and his team [23] trained a GAN to generate synthetic brain tumor MRIs and evaluated the performance by a segmentation network subsequently trained with the generated MRI data. Kossen and others [16] successfully generated anonymized and labeled TOF-MRA patches that retained generalizable information and showed good performance for vessel segmentation.

In the fields of the generative adversarial network, the goal of image translation is to translate an input image from one domain to another by using given input-output image pairs as training data. Current state-of-the-art approaches [14,15,24,25,26] typically employ conditional GANs to optimize the networks either with explicitly labeled paired data or by forcing cyclic consistency with unpaired data. In contrast to L1 loss or MSE loss, the adversarial loss has become a popular choice for many image-to-image tasks and can often produce higher image quality. The discriminator is like a trainable loss function which can be automatically adapted to the differences between the generated and real images in the target domain while training. Image translation is one of the most promising applications in computer vision.

The pix2pix [26] proposed by Isola and his team in 2017 is a common conditional GAN framework for image translation. In this framework, the goal of the generator G is to translate the semantic label graph into a real image, while the goal of the discriminator D is to discriminate between the real image and the translated image. The framework is a supervised learning neural network based on the U-Net architecture. Following this framework, Wang and others [14] proposed a coarse-to-fine generation method in 2018 called pix2pixHD for synthesizing higher resolution images from semantic label graphs. The method first learns low-resolution translations, and then gradually scales up to high-resolution. The quality of synthesis images is effectively improved by applying new adversarial loss, multiscale generator, and discriminator architecture. To reduce the heavy computational burden of convolution of high-resolution feature maps, Liang and others [27] proposed a Laplacian Pyramid Translation Network (LPTN) in 2021 to perform the two tasks of attribute and content detail transformation simultaneously.

### 2.2. Datasets

The data used in this paper were obtained from Shanghai Jiao Tong University Affiliated Sixth People’s Hospital specializing in thyroid disorders in Shanghai, China. The original size of all ultrasound images was 800 × 555 (in Figure 1a). We collected a total of 726 thyroid ultrasound elastography images with corresponding conventional images (pressed B model ultrasound images obtained when elastography was performed) from 397 patients between 2019 and 2021. The patients included 98 males and 299 females, with a minimum age of 20 and a maximum age of 82, and a mean age of 43.6 (standard deviation of 13.4). Under the specialists’ recommendations, thyroid nodules with sizes between 5 and 30 mm were included, while nodules with coarse calcification and predominantly cystic nodules were excluded. To enhance the model generalization capability among different types of equipment, only simple RGB images in the range of 0–255 are used. The training set has 646 images and the testing set has 80 images. This study was approved by the ethics committee of Shanghai Jiao Tong University Affiliated Sixth People’s Hospital.

### 2.3. Network Architecture

To generate strain elastography images from conventional ultrasound images, three key requirements have to be considered: (1) The original input data does not just contain input ultrasound images. They are the focused regions of interest (ROI) manually circled by the specialist (the green dash rectangle in Figure 1) to generate the corresponding elastography images. (2) The ultrasound images are grayscale images, which contain less information. The information of focused features such as nodule locations and tissues need to be extracted to generate elastography images. (3) The elastography image mainly relies on color to distinguish tissue elasticity (shown in Figure 1). Therefore, it is important to consider color information in the generation process, the network structure, and loss function.

Based on the above considerations, we designed a framework (see Figure 2) to generate elastography images from conventional ultrasound images by first preprocessing and then using the AUE-net (see Figure 3). As shown in Figure 1a, the original data are not a simple ultrasound image-elastography image pair, but only a green box manually circled by a specialist to mark the focus area, and the green box is not uniform in color, size, and position. Therefore, we coiffed the images by an image gradient algorithm.

First, we cropped the raw image at fixed locations (left, right, upper, and lower boundaries) in the image according to different ultrasound devices:(1)$$I_{flc}=\mathrm{FLC}\left(I_{ori}\right),$$
where $I_{ori}$ and $I_{flc}$ are original images and fixed locations cropped images respectively. FLC(·) is the fixed locations cropping operation. Then, the image obtained from the first crop (Figure 1b) is traversed in the vertical direction in row units, where the number of points with nongray pixels is counted within each row. After that, the gradients of the data in all rows are calculated and the two largest gradients are selected as the upper and lower boundaries of the ROI region. Finally, the above operation is repeated in the horizontal direction to obtain the left and right boundaries of the ROI area.
(2)$$I_{ROI}=\mathrm{CBC}\left(I_{flc},\underset{x}{\mathrm{argmax}}\nabla^{1}M_{color}\right),$$
where $I_{ROI}$ and $I_{flc}$ are output ROI images and fixed locations cropped images respectively. CBC(·) is the color boundary cropping operation. The matrix $M_{color}$ contains the number of color pixels. *x* is a boundary position of ROI. We obtained the final cropped data by the above algorithm (Figure 1c).

As shown in Figure 3, the AUE-net consists of two main components: a generator and a discriminator (omitted in Figure 3). The generator and discriminator have several submodules, including the nodule position attention module (AUE-SA), the AUE residual block (AUE-ResBlk), and the color attention module (AUE-CA), as shown in Figure 4. We will discuss them separately in this section.

(1) AUE-ResBlk: our AUE residual module was inspired by the work of Spatially Adaptive Denormalization (SPADE). However, unlike SPADE using semantic masks as inputs, we extended it to the underlying module where global features are extracted. We took the previously extracted feature maps as input and fused them with the newly extracted features (see Figure 4c)):(3)$$OUT=\gamma_{i}\left(PF\right)\cdot \frac{{pf}_{i}-\mu_{i}}{\sigma_{i}}+\beta_{i}\left(PF\right),$$
where PF are previously extracted feature maps, $pf_{i}$ denote the activations of the *i*th layer of a deep convolutional network. $\mu_{i}$ and $\sigma_{i}$ are the mean and standard deviation of the $pf_{i}$ activations. $\gamma_{i}$ and $beta_{i}$ denote the conversion of PF to the scaling and bias values. Through this approach, it is possible to make full use of the organizational feature information extracted by the upper layer network.

(2) Generator: our coarse-to-fine image generator was based on the architecture proposed by Johnson and others, which has proven to be successful for neural style migration for images up to 512 × 512. We first analyzed the characteristics of the elastography image generation task. And combining with these, we added spatial attention (AUE-SA) module (see Figure 4b) to find the nodal positions at the front end, adding an AUE residual block before the up-sampling module for better feature fusing, and adding a channel attention (AUE-CA) module (see Figure 4a) for analyzing the color distribution at the end. However, our model demonstrated that the AUE-CA module will better generate elastography images by learning to reasonably assign the weights of the red, blue, and yellow channels at the final output, while the AUE-SA module can learn the tissue location features in the input ultrasound images to better extract features and generate elastography images of the target region (i.e., thyroid nodules). The generator can also be described by common GAN:(4)$$\min_{G}\max_{D}L_{GAN}\left(G,D\right),$$
where the loss function $L_{GAN}\left(G,D\right)$ is:(5)$$E_{\left(s,x\right)}\left[\log D\left(s,x\right)\right]+E_{x}[\log\left(1-D\left(G\left(x\right),x\right)\right],$$
where *G* for generator, *D* for discriminator, *x* for inputs, and *s* for labels.

(3) Discriminator: a multiscale, patch-based discriminator with the InstanceNorm (IN) [28] was utilized from the pix2pixHD method (omitted in Figure 3). Furthermore, we applied the Spectral Normalization [29] to all the convolutional layers of the discriminator. And to enhance the nonlinear mapping capability, we used the Leaky-ReLU activation function instead of the ReLU activation function. At different scales, the discriminator can act as a feature extractor for the generator to optimize feature matching losses. These discriminators were trained to use authentic and synthetic images at different scales.

(4) Color-Loss: since elastography images use color to represent different hardness, generating a more natural elastography image means developing an image with a more realistic color distribution. To obtain a better color distribution of the elastography images, a loss function that measures the color difference between the generated image and the real image is needed. We need to eliminate the effect of texture and content in the image and measure only the differences in brightness, contrast, and primary colors in the image. Hence, we processed the image with Gaussian blur to ignore the small range of pixel differences by an additional convolution layer, and then we computed the distance between the obtained feature maps to express the color differences. This color loss between X and Y could be written as:(6)$$L_{\mathrm{color}}\left(X,Y\right)=\mathrm{dis}\left(X_{b},Y_{b}\right),$$
where dis(·) denotes the function for computing the distance. After comparing the actual experimental results by using different distance functions, the sum of Euclidean distance and L1 distance are used. $X_{b}\;$ and $Y_{b}$ denote the two images after Gaussian blur respectively, for example:(7)$$X_{b}\left(i,j\right)=\sum_{k,l}X\left(i+k,j+l\right)\cdot G\left(k,l\right),$$
where G(k,l) denotes the Gaussian kernel with the size of k×l. Our full GAN loss combined with the generator loss $L_{GAN}$, feature matching loss $L_{FM}$ and color loss $L_{color}$ is as follows:(8)$$\min_{G}\left(\left(\max_{D_{i}\in D}\sum L_{GAN}\left(G,D_{i}\right)+\beta \cdot L_{\mathrm{color}\;}\left(G,D_{i}\right)\right)+\alpha \cdot \sum L_{FM}\left(G,D_{i}\right)\right),$$
where α controls the weight of the feature loss and β presents the importance of color loss. $D_{i}$ is the result of *i*th discriminator.

In summary, AUE-net fully considers and leverages the characteristics of the input ultrasound image and the elastography image generation task. It is implemented and optimized by adding a nodal position attention module, a color channel attention module, an ultrasound feature fusion module, and a loss function optimization to optimize color distribution.

### 2.4. Implementation Details

All experiments are conducted in the following computer environment as shown in Table 1.

We trained a total of 1500 epochs using the batch size of 8. The number of down-sampling modules and up-sampling modules were both three. Each module had a convolutional layer (deconvolutional layer in the up-sampling module), a normalization layer and an activation layer. The residual module consist of nine blocks. These values were chosen after performing a preliminary grid-search-based optimization procedure. Due to the specificity of dynamic games in the GAN training, the quality of generated images could not be simply estimated by the loss function [30]. According to specialists, we selected 1500 epochs as the best result from the generation quality of multiple training epochs such as 500, 1000 and 1500 (in Figure 5). The perceptual loss [24] consists of feature maps of the Relu1_1, Relu2_1, Relu3_1, Relu4_1, Relu5_1 layers of the pretrained VGG-19 model with weights [1/32,1/16,1/8,1/4,1]. Spectral normalization [29] was applied to the whole network. We utilized IN [28] as both generator and multiscale discriminator. The Adam [31] optimizer was used with two time-scale update rules with setting $\beta_{1}=0.5$ and $\beta_{2}=0.999$. The learning rates for the generator and discriminator were 0.0002 and 0.0001, respectively.

To enhance the robustness and generalization of the model, there is a certain probability (e.g., 0.5) that different data augmentation operations will be performed at input data, including ±20 pixels of horizontal, vertical, or diagonal translation, horizontal and vertical mirror inversion, and $\pm 15^{\circ }$ of rotation.

Since the elastography image generated from each ultrasound image needs to be reproducible, we compared our method with paired I2IT methods using conditional GANs without a random noise map input, i.e., pix2pix [25], pix2pixHD [14], and LPTN [27] (in Table 2).

### 2.5. Evaluation Metrics

In terms of evaluation metrics, we used traditional image similarity metrics for evaluation, such as peak signal-to-noise ratio (PSNR) [32] and structural similarity index measure (SSIM) [33]:(9)$$\begin{array}{cc} \; & \mathrm{PSNR}\left(I,G\right)=\frac{10}{C}\sum \log_{10}\left(\frac{MAX_{I}^{2}}{\frac{1}{mn}{∥I-G∥}_{2}^{2}}\right), \end{array}$$(10)$$\begin{array}{cc} \; & \mathrm{SSIM}\left(I,G\right)=\frac{\left(2\mu_{I}\mu_{G}+c_{1}\right)\left(2\sigma_{IG}+c_{2}\right)}{\left(\mu_{I}^{2}+\mu_{G}^{2}+c_{1}\right)\left(\sigma_{I}^{2}+\sigma_{G}^{2}+c_{2}\right)}, \end{array}$$
where *I* and *G* are two images with m×n, *C* is the number of channels. ${MAX}_{I}^{2}$ is the maximum pixel value of the image which is 255 here. ${∥\cdot ∥}_{2}$ stands for the Euclidean norm. μ is the mean. $\sigma^{2}$ is the variance. $\sigma_{IG}$ the covariance of *I* and *G*. $c_{1}$ and $c_{2}$ are two variables to stabilize the division with weak denominator. The paper also uses the Frechet Inception Distance (FID), first proposed by Heusel and others [34]:(11)$$\mathrm{FID}\left(I,G\right)={∥\mu_{I}-\mu_{G}∥}_{2}^{2}+\mathrm{tr}\left(C_{I}+C_{G}-2\sqrt{C_{I}C_{G}}\right),$$
where μ is the mean, tr is the trace, and *C* is the covariance matrix. The FID utilizes a pretrained Inception network on the ImageNet dataset to evaluate the quality of the GAN-generated images. The generated and real images are first fed into the pretrained Inception network. Then, the activation means of the final layer and the covariance of the assumed Gaussian distribution are extracted. Finally, the Frechet distance between them is calculated. The FID is computed over the learned feature space and was shown to correlate well with human visual perception [35].

Finally, to measure the clinical value of our method, we asked thyroid specialists to provide scores. First, two specialists scored randomly disrupted real and generated images according to the Ueno–Rago criterion [36,37]. Then two scores of the same image were matched, with the same being 1 and different being 0. The scoring accuracy of the entire test set was tallied. This index is consistent with the clinical prediction process for thyroid nodules and can be applied for clinical use.

## 3. Results

### 3.1. Visual Comparisons

We compared the visual performance of the elastography image generated by our model with that of three networks (pix2pix, pix2pixHD, and LPTN). As shown in Figure 6, the AUE-net proposed in this paper outperformed all other three methods in terms of imaging realism and quality.

Specifically, AUE-net translated the input strained ultrasound image into the corresponding elastography image with little texture distortion (Figure 6). The results showed a more realistic and natural color distribution, and the masked part of the original image without ultrasound information was partially removed. Among these methods, pix2pixHD was the second-best network. However, it introduced more hardened features to the nodule locations, i.e., the blue area of the nodule, and over-representation of hardened features, due to insufficient reconstruction capability of the decoder for the nodule. In contrast, AUE-net enhanced the hardened information generation in the nodule region during the encoding-decoding process by the spatial attention module and the channel attention module, producing results closer to the actual image.

In general, most of the existing I2IT methods are based on the autoencoder generation framework (LPTN is based on the Laplace pyramid) with three main modules: (1) decomposing content information and attributes on the low-dimensional latent space by the encoding process; (2) transforming the attributes within the latent space by the residual module; (3) reconstructing the image from the transformed attributes by the decoding process. The ability to reconstruct elastography images was simulated by the network parameters of the encoder-decoder, but the characteristics of elastography images were not taken into account in these methods, which led to mediocre results. Our proposed AUE-net was an encoder-decoder network that took into account the characteristics of nodule position correlation and color distribution correlation of elastography images. Better results could be generated through the spatial and channel attention modules. The elastography images generated from the perspective of image algorithms were more nodule location correlated and could assist specialists with nodule localization. The image algorithm helped to reduce human efforts and decreased the need for equipment.

### 3.2. Quantitative Results

This subsection compared the AUE-net with the I2IT methods described above using three image similarity metrics (PSNR, SSIM and FID) and scores from the specialists. The results of our experiments in terms of image quality were shown in Table 3.

As shown in Table 3 and Table 4, the results generated by our model were the best among the four models. The mean accuracy of scores given by two specialists was 84.38%. The pix2pixHD achieved an accuracy with 75.4%. These results demonstrated that the AUE-net proposed in this study showed better performance for generating elastography image than the general I2IT methods.

Based on specialists’ opinions, the elastography images generated by our model could meet the needs of clinical diagnostic applications and provide practical value. We reviewed the cases that showed errors from the specialists. Specifically, the scoring accuracy was 66.67% for grade I, 78.57% for grade II, 82.61% for grade III, 90.48% for grade IV, and 100.00% for grade V. The lower accuracy of grade I was probably because of its small number. Most of the errors were between grade II and grade III. These errors were due to visual observation. In the Ueno–Rago criterion [36,37], grade II is with green predominating, while grade III is with blue dominating. It is therefore quite a challenge to determine accurately when the lesion area is mixed with blue and green.

There were also a few other error cases that seem to result from unknown circumstances. For example, in Figure 7, the true elastography image showed that most tissue was hardened. There could be two possibilities: either the device was biased which resulted in incorrect calculation or most of the targeted tissue became hardened. Special cases like this should be excluded when assessing the validity and performance of our model.

Although this paper showed promising results in terms of image quality and scoring accuracy, our current study was limited in several aspects. First, due to the inherent difference between image algorithms and physical methods, the proposed AUE-net could not generate the same strain elastography images as the actual images. Our method would not be used as a substitute for the strain elastography technique, but was considered as a prefractionated reference for tissue. Second, our method had not yet investigated generalization among different devices due to the significant variation of elastography images produced.

## 4. Conclusions

With strain elastography becoming a promising new tool that can effectively improve diagnostic accuracy for thyroid nodules, we built and proposed in this paper a new method for generating elastography images based on the characteristics of thyroid elastography images. The AUE-net method used a base pix2pix architecture with a fused residual module and combined with a spatial attention module, a channel attention module, and a color loss feature. Results from quantitative and qualitative evaluation showed that the elastography images generated by presented model produced better quality compared to that of other reported models. Accordingly, conventional ultrasound image harbors information that can be linked to tissue elasticity. However, our work has some limitations. Since the current study used postpressure ultrasound images to generate the corresponding elastography images, the experience of specialists was still required. For our next step, we will continue to study the generation of elastography images by using direct prediction of plain ultrasound images to eliminate the influence of human factors.

## Figures and Tables

**Figure 1 diagnostics-12-00253-f001:**
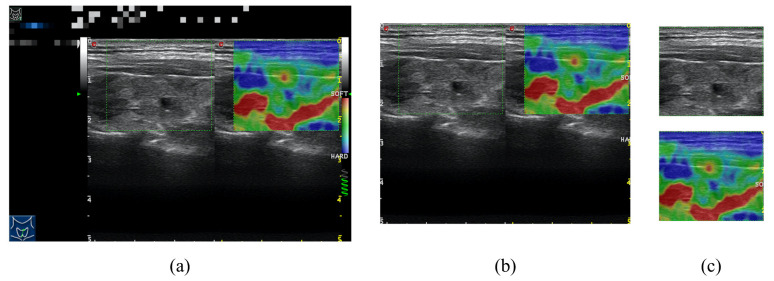
Example image of three steps in data preprocessing. (**a**) Original images obtained from same ultrasound equipment, which have same dimensions of 800 × 555. (**b**) First cropped image using fixed positions according to equipment information. (**c**) Final cropped ultrasound image-elastography image pair obtained by color gradient cropping algorithm.

**Figure 2 diagnostics-12-00253-f002:**
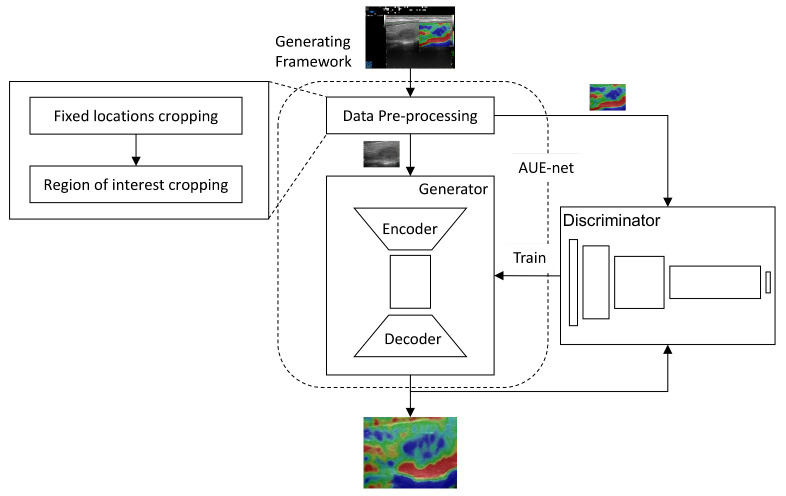
Whole flow of our generation framework.

**Figure 3 diagnostics-12-00253-f003:**
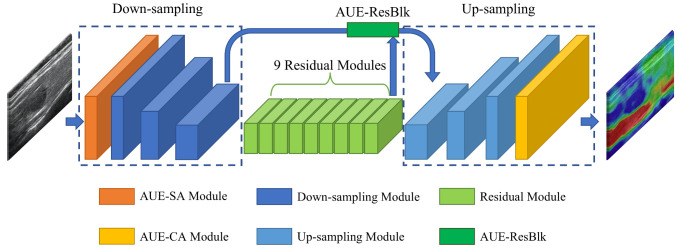
Architecture of AUE-net, which consists of 1 AUE-SA module, three down-sampling modules, nine residual blocks, three up-sampling modules, one AUE-CA module, and one AUE-ResBlk.

**Figure 4 diagnostics-12-00253-f004:**
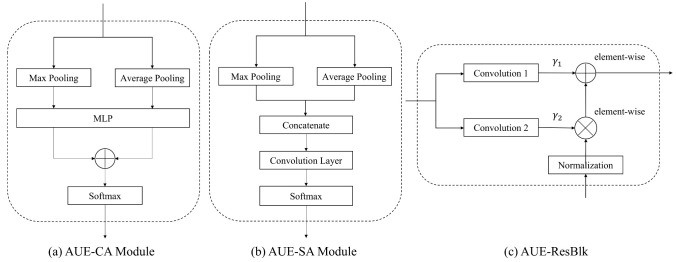
Submodules of our AUE-net. (**a**) is used to create a color attention map for elastography image color. (**b**) helps to locate nodule in elastography image. (**c**) helps making full use of feature information extracted before.

**Figure 5 diagnostics-12-00253-f005:**
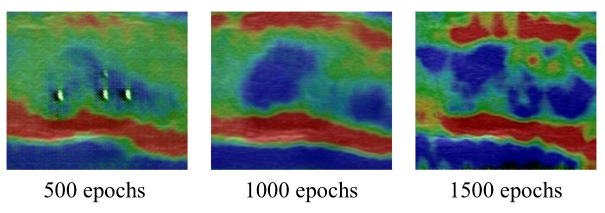
Generation results after 500, 1000, and 1500 epochs.

**Figure 6 diagnostics-12-00253-f006:**
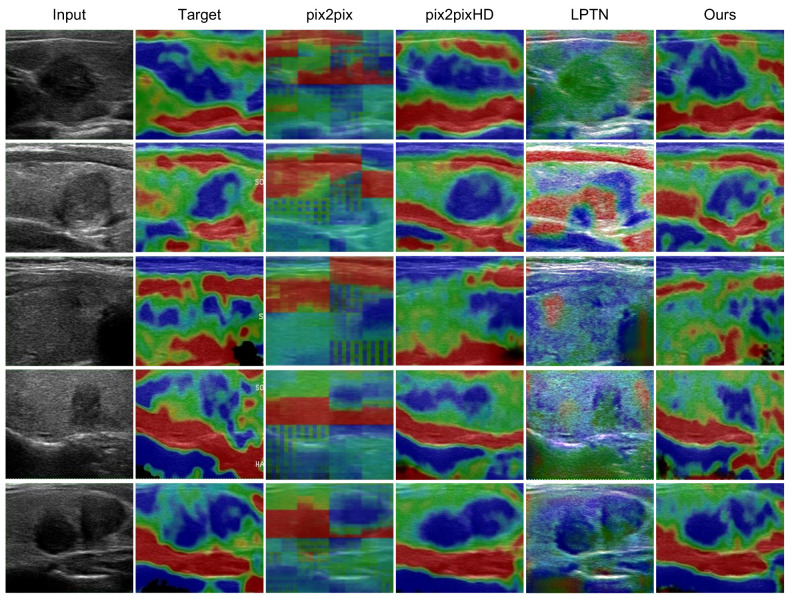
Generation results compared to pix2pix, pix2pixHD and LPTN. First two columns show input ultrasound data and target elastography image.

**Figure 7 diagnostics-12-00253-f007:**
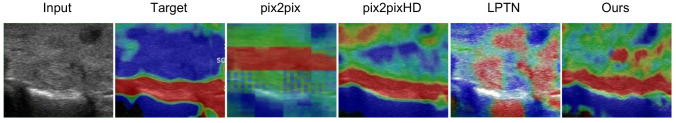
Special example in test images shows that most tissue is hardened, while generated result is not good enough.

**Table 1 diagnostics-12-00253-t001:** Experimental environment information.

Items	Information
Operating System	Linux Red Hat 4.8.5-36
CPU	IBM(R) POWER9 (3.8 GHz)
GPU	NVIDIA Tesla V100 32 G

**Table 2 diagnostics-12-00253-t002:** Comparison among our method and state-of-the-art methods.

Methods	Year	Highlights
pix2pix [25]	2017	General-purpose solution to image-to-image translation based on cGAN
pix2pixHD [14]	2018	Synthesizing high-resolution photo-realistic images A novel adversarial loss, new multiscale generator, and discriminator architectures
LPTN [27]	2021	Laplacian pyramid decomposition and reconstruction Speeding-up the high-resolution photo-realistic I2IT tasks
AUE-net	2021	AUE attention modules, AUE residual blocks and color loss Generating strain elastography images from conventional ultrasound images

**Table 3 diagnostics-12-00253-t003:** PSNR, SSIM, FID, and scoring accuracy of our method compared to that of state-of-the-art methods.

Methods	PSNR ↑	SSIM ↑	FID ↓	Score Acc ↑
pix2pix [25]	12.688	0.281	205.38	30.8%
pix2pixHD [14]	28.651	0.456	56.71	75.4%
LPTN [27]	10.879	0.364	155.24	32.3%
Ours	28.736	0.499	51.09	84.4%

↑ and ↓ denote higher is better and lower is better, respectively.

**Table 4 diagnostics-12-00253-t004:** Accuracy of scores from two specialists.

	Correct Amount of 80 ↑	Score Accuracy ↑
Specialist 1	67	83.75%
Specialist 2	68	85.00%
Mean	67.5	84.38%

↑ and ↓ denote higher is better and lower is better, respectively.

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
