# Peer review of "AUE-Net: Automated Generation of Ultrasound Elastography Using Generative Adversarial Network"

_diagnostics, 2022, doi:10.3390/diagnostics12020253_

Round 1
Reviewer 1 Report
- The main contribution of the research is not getting clear from the introduction part.
- The overall organization of the paper must be presented in the last paragraph of the introduction section.
- All the parameters used in eq 1-8 must be elaborated in the text.
- In section 2.4 authors stated 1500 epoch training, why the authors have not considered early stopping to avoid overtraining neural networks?
- In table 2, its better to provide the citation of the papers compared with the proposed work, so that it will be getting easy for readers to follow.
Author Response
Dear Reviewer,
Thanks very much for taking your time to review this manuscript. We thank you for the time and effort that you have put into reviewing the previous version of the manuscript. The suggestions have enabled us to improve our work. Please find our point-by-point responses in the attachment and our revisions/corrections in the re-submitted files. Thanks again!

Reviewer 2 Report
>> The language usage throughout this paper need to be improved, the author should do some proofreading on it. Give the article a mild language revision to get rid of few complex sentences that hinder readability and eradicate typo errors.
>> Your abstract does not highlight the specifics of your research or findings. Rewrite the Abstract section to be more meaningful. I suggest to be Problem, Aim, Methods, Results, and Conclusion.
>> Need to highlight the main findings (brief) in the abstract.
>> The proposed method and experiments are not clearly illustrated.
>> Add main contributions list as points in the Introduction section.
>> Add the rest organization section at the end of the Introduction section.
>> The authors should consider more recent research done in the field of their study Such as:
1) Hybridizing Convolutional Neural Network for Classification of Lung Diseases. International Journal of Swarm Intelligence Research
2) Neural network and multi-fractal dimension features for breast cancer classification from ultrasound images. Computers & Electrical Engineering
3) Fully‐automatic identification of gynaecological abnormality using a new adaptive frequency filter and histogram of oriented gradients (HOG). Expert Systems
4) Diagnosing pilgrimage common diseases by interactive multimedia courseware. Baghdad Science Journal
5) Fully Automatic Segmentation of Gynaecological Abnormality Using a New Viola–Jones Model. Computers, Materials & Continua
>> Redraw Figure 2. The whole flow of our generation framework. I suggest to redraw with more specific steps and description.
>> I feel that more explanation would be need on how the proposed method is performed.
>> If no one has proposed before a method like the proposed algorithm, this claim should be highlighted much more. Else, it should be indicated who has done this, and it should be indicated what the innovations of the current paper are.
>> Authors should add the parameters of the algorithms.
>> A comparison with state of art in the form of table should be added
>> Results need more explanations. Additional analysis is required at each experiment to show the its main purpose.
>> The Limitations of the proposed study need to be discussed before conclusion.
Author Response

(The authors gave the same response as above.)

Round 2
Reviewer 1 Report
- In table 2 the proposed work must be presented also.
Author Response
Dear Reviewer,
Thanks very much for taking the time to review this manuscript. We thank you for the time and effort that you have put into reviewing the previous version of the manuscript. The suggestions have enabled us to improve our work. Please find our point-by-point responses in the following and our revisions/corrections in the re-submitted files. Thanks again!
Point 1: In table 2 the proposed work must be presented also.
Response 1: We have added the information of the proposed work in Table 2.